# Endoscopic Guided Dilations without Intralesional Corticosteroid Injections: Pediatric Crohn’s Patients Case Series

**DOI:** 10.3390/reports7040081

**Published:** 2024-09-24

**Authors:** Leo Fawaz, Yousif Slim, Peter N. Freswick

**Affiliations:** 1Pediatric Gastroenterology Fellowship—Children’s Hospital of Michigan, Detroit, MI 48201, USA; 2Family Medicine Residency—Wayne State University, Detroit, MI 48201, USA; 3Division of Pediatric Gastroenterology—Helen DeVos Children’s Hospital, Grand Rapids, MI 49503, USA

**Keywords:** Crohn’s, dilations, strictures

## Abstract

**Background and Clinical Significance:** The treatment for pediatric Crohn’s disease (CD) has shifted over the years from steroids and immunomodulators to biologics with the goal of histological and clinical remission. Endoscopic balloon dilation (EBD) has been utilized for stricturing disease, even in the pediatric population. EBD has been shown to be effective and minimally invasive, though historically, has been performed on patients with persistent mucosal inflammation. As such, intralesional corticosteroid (ILC) injections have been traditionally utilized during EBD. However, intralesional corticosteroid efficacy among pediatrics patients in deep endoscopic remission is unknown. **Case Presentation:** We report four patients that demonstrated at least initial successful dilations without intralesional steroid injections. **Conclusions:** The use of ILC injections during routine EBDs in pediatric patients should be further explored in randomized control trials.

## 1. Introduction and Clinical Significance

Pediatric patients with Crohn’s disease (CD) commonly present with right lower quadrant abdominal pain and non-bloody diarrhea. CD affects any part of the gastrointestinal tract from the mouth to the anus, with skip lesions at the terminal ileum (TI) [1]. The incidence of Crohn’s disease in children continues to increase with an incidence of 2.5 to 11.4 per 100,000 and a prevalence of 58 per 100,000 [2]. Areas of inflammation occasionally develop stricturing disease, most commonly in the TI and ileocecal region [3]. Achieving and maintaining clinical remission is the primary goal of therapy in pediatric patients with a treatment plan based on disease location and inflammatory activity [1].

While biologic therapy has improved CD outcomes, stricturing disease remains an unfortunate reality for many patients. Strictures are a common manifestation of CD, and therapeutic management based on location and severity has recently changed [3]. Providers have utilized endoscopic balloon dilation (EBD) with ILC for therapeutic stricture management, as complication rates are lower in EBD procedures versus surgical intervention, and short-term results document relief from abdominal pain, nausea, vomiting, and abdominal distension [4]. EBD has also been used to avoid surgery in many pediatric patients, with only 30–50% of patients who undergo EBD eventually requiring surgery [4]. Although the literature is sparse in the pediatric population, a recent study in 2021 concluded that patients who underwent EBD with intralesional corticosteroids (ILC) had no procedural complications and did not require any further surgical intervention during follow-up periods [5]. As such, ILC injections have been traditionally utilized during EBD. However, ILC efficacy among pediatric patients in deep endoscopic remission is unknown. Two randomized placebo control studies have been reported, with conflicting results. In a study by East et al., 13 patients aged 18 and older were randomized to receive steroid injections (*n* = 7) versus placebo treatment. Results showed that 1 of the 6 in the placebo group and 5 out of 7 in the steroid group required re-dilation. This raised the question of whether triamcinolone injection trends towards a worse outcome [6]. Di Nardo et al. also randomized patients with short colonic strictures (<5 cm) to receive steroids or saline, with 14 out of 15 in the intervention group showcasing successful results [7]. Thus, it was concluded that ILC injections after EBD was an effective treatment strategy for decreasing the need for re-dilation. However, complications have been reported with ILC injections that include perforation, delay in linear growth, or intramural infection [8]. Lastly, there is a large variation between patient groups, concurrent medications and techniques utilized in existing literature that trends towards ILC injections to support the routine use [9]. Based on the limited and contradictory efficacy data supporting the use of ILC in pediatrics and the potential side effects, we elected to proceed with dilations without steroid injections.

We present four pediatric patients with underlying CD, differing in clinical presentation and symptoms. Our pediatric patients were treated with biologics (adalimumab, infliximab, ustekinumab; sometimes in combination with immunomodulator therapy) and chosen for stricture dilation after patients demonstrated mucosal remission (no inflammation). All strictures were <5 cm in length, non-angulated, fibrotic and non-ulcerated. No active inflammation was present at the time of dilations. None of our patients received ILC injections. Dilations were targeted to a goal of 20 mm with planned repeat scope approximately one year later to assess stricture characteristics. Once the goal was achieved, no repeat scopes were performed. All dilations utilized a colonic 5.5 cm long balloon dilator. No wire or fluoroscopy-guided dilatations were performed. All dilations were held for 60 s after drift cessation. Stricture size was endoscopically assessed, then dilated by 1 to 1.5 mm increments until superficial mucosal tearing was visualized.

Patients were excluded if stricture length was >5 cm, unable to be dilated (stricture diameter less than 4 mm), failure of steroid-free deep remission on biologic and/or combo biologic/modulator therapy, and family refusal with desire to proceed with surgical intervention over likely repeated colonoscopies.

## 2. Case Presentation

Patient 1 is a 17-year-old female who presented in January 2018 with cyclical episodes of vomiting and right upper quadrant/right lower quadrant abdominal pain, and weight loss. Magnetic resonance enterography (MRE) confirmed a 7 cm thickening of the TI with a proximally dilated small bowel. Strictures were located at the distal and more proximal TI. She started treatment with Infliximab 5 mg/kg every 8 weeks. She received dilations in November 2019 of the distal TI from 12 to 18 mm and of the proximal TI to 16.5 mm. She was then switched to adalimumab at 40 mg and later, the dose escalated to 80 mg in 2020. Both strictures in November 2020 were dilated to 20 mm, with biopsies showing histological remission. Severe disease re-emerged in November 2022 with C. difficile infection, ultimately ending in bowel perforation, and she was switched to ustekinumab. In January 2023, she was readmitted with either perforated appendicitis or ileal perforation and completed an antibiotic course. The patient underwent a laparoscopic-assisted ileocecectomy in March 2023.

Patient 2 is a 16-year-old male who presented in July 2019 with five days of severe RLQ abdominal pain and peritoneal signs requiring urgent admission to the hospital. The initial CT showed a right lower quadrant abscess and inflammation, and he was started on broad spectrum antibiotics. An MRE showed inflammation of the ileum with skip lesions. July 2019, colonoscopy revealed cecal, ileocecal valve (ICV), and proximal right colon ulceration with inability to intubate the TI. A biopsy showed mild chronic active colitis. Infliximab 5 mg/kg every 8 weeks was then started. In August 2019, a repeat colonoscopy showed a colonic stricture between the cecum and the proximal right colon, and an ICV stricture. A repeat scope in July 2020 showed a normal colon, and an ICV stricture dilated to 11 mm from the previous 8 mm, still with inability to intubate TI. In December 2020 and August 2021, the ICV underwent dilation from a previous 9 mm to 18 mm and then 12 mm to 18 mm, respectively. Biopsies showed mild, chronic active ileitis. Infliximab increased to 10 mg/kg every 8 weeks. The last dilation in June 2022 showed ICV dilation from 13.5 to 20 mm, with moderate renting. No repeat dilations have been needed to date.

Patient 3 is an 18-year-old male who presented in February 2015 with hematochezia, diarrhea, and abdominal pain. An EGD/colonoscopy in 2015 showed CD of the stomach, TI, and large bowel. The patient’s MRE showed moderate active inflammatory changes involving the sigmoid colon, distal descending colon, and TI, with suboptimal bowel distention. He started on 6-mercaptopurine at 25 mg three times daily, and steroids and allopurinol at 50 mg daily. A repeat MRE in February 2020 showed 10–15 cm sigmoid colitis. He then started on Rowasa enemas daily. The fecal calprotectin level in September 2020 was 419. In February 2021, he underwent dilation of a new stricture of the proximal ascending colon from 7 mm to 15 mm, with moderate renting. Therapy was changed to adalimumab at 40 mg weekly and methotrexate at 25 mg. Then, in August 2021, the patient had another dilation from 15 mm to 20 mm, no renting noted. The dilation progression is shown in Figure 1, Figure 2, Figure 3, Figure 4 and Figure 5. Biopsies showed mild, chronic ileitis, with eosinophilia and mild chronic colitis. A fecal calprotectin level repeated in January 2022 was 50. No repeat dilations have been required to date and he has only had mild GI symptoms since then.

Patient 4 is a 15-year-old male, who presented with severe weight loss, and abdominal pain. He underwent an EGD/colonoscopy in November 2021 showing cecal and ICV ulceration, with inability to intubate the TI. His biopsies showed mild chronic cecitis with erosion and mild activity with moderate crypt distortion. He started adalimumab at 40 mg every two weeks. In January of 2022, his symptoms returned and an MRE showed a 6–10 cm terminal ileum stricture with proximal moderate dilation and collapsed cecum. He was changed to adalimumab at 40 mg weekly. In June 2022, the fecal calprotectin level was >2500. In September of 2022, a colonoscopy showed a stricture in the ICV, dilated from 4 mm to 13.5 mm, with a food bezoar removed from the TI. Then, in November 2022, the ICV dilated from 8 to 15 mm; subsequently, a methotrexate 25 mg injection was started. Again, he underwent dilation of ICV in December 2022, from 12 to 15 mm, with mild renting. In May 2023, the colon and TI were normal, but he underwent dilation of the ICV from 14 to 18, with moderate renting. The last dilation in September 2023, with the ICV dilated from 18 to 20, with mild renting. Biopsies last showed chronic ileocecitis, non-active.

## 3. Discussion

IBD treatment has changed drastically in the past decade; treatment options for CD have transitioned to more tolerable biologics (adalimumab/infliximab), and increasingly efficacious biologic options are available for patients. We report four patients with endoscopic and symptomatic remission after EBD without ILC, albeit one patient only transiently improved and eventually required surgery. Patients aged 15 to 18 years old had differing clinical signs and symptoms. Patient characteristics are discussed in Table 1 and Table 2. Each patient was dilated to 20 mm with at least one follow-up endoscopy to ensure remission.

Although some literature supports using ILC to decrease repeated surgeries, our cases were not subject to injections during dilations. Our patients did require repeat dilations, with slight re-stricturing at times prior to the next dilation. However, three of the four patients were still able to maintain successful dilations. In addition, there have been numerous randomized control studies in adults showcasing success of EBD for strictures without ILC. A study on intestinal strictures in patients with IBD by Won Lee et al. demonstrated successful dilations of 86% of adult patients (26/30) on the first attempt and 100% (30/30) on the second dilation without ILC [10]. Hirdes et al. constructed a double-blind study that included 60 adult patients with esophagogastric anastomotic strictures randomized to receive either triamcinolone or saline and then were dilated to 16 mm. Overall, it was found that triamcinolone did not statically decrease the frequency of repeated dilations or of the dysphagia free periods in the patients [11]. Lastly, Atreja et al. documented 128 patients with Crohn’s disease who underwent endoscopic stricture dilation, and compared the need for reintervention or surgery with and without the use of ILC for both primary and anastomotic strictures. The authors concluded that ILC did not decrease the risk of surgery or reintervention [12]. Although there are many randomized trials for adults with IBD showcasing no difference between the usage, there continues to be limited numbers in pediatrics.

Limitations of the case series include a lack of a comparison group, small sample size, and inability to generalize to a larger population. Future controlled trials should compare the risk and benefits of dilations with and without intralesional steroids.

## 4. Conclusions

This case series showcased three patients with successful EBD without intralesional steroids. However, given the limitations of a case series, the use of ILC injections during routine EBD in pediatric patients should be further explored in randomized control trials.

## Figures and Tables

**Figure 1 reports-07-00081-f001:**
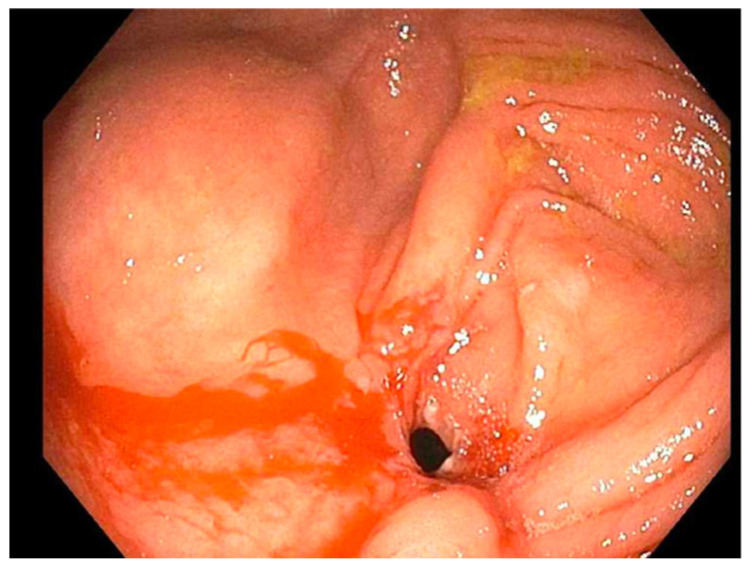
Patient 3 initial stricture of right colon (2020).

**Figure 2 reports-07-00081-f002:**
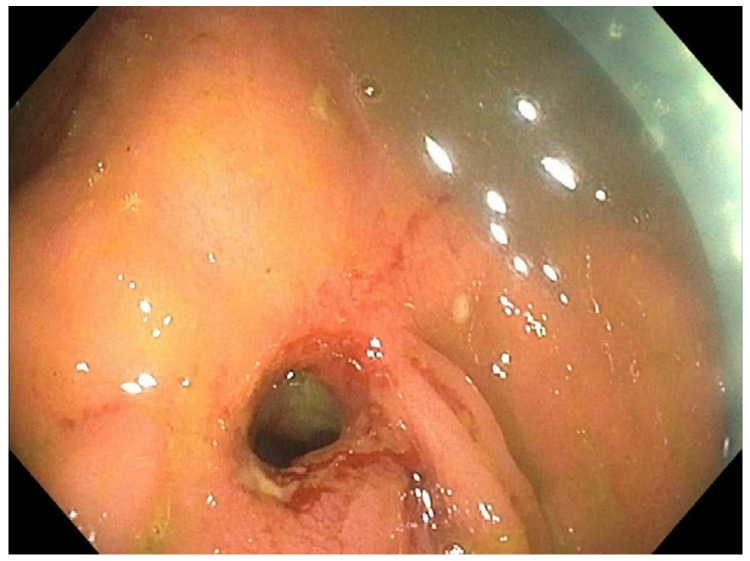
Patient 3 February 2021; 7 mm stricture pre-dilation.

**Figure 3 reports-07-00081-f003:**
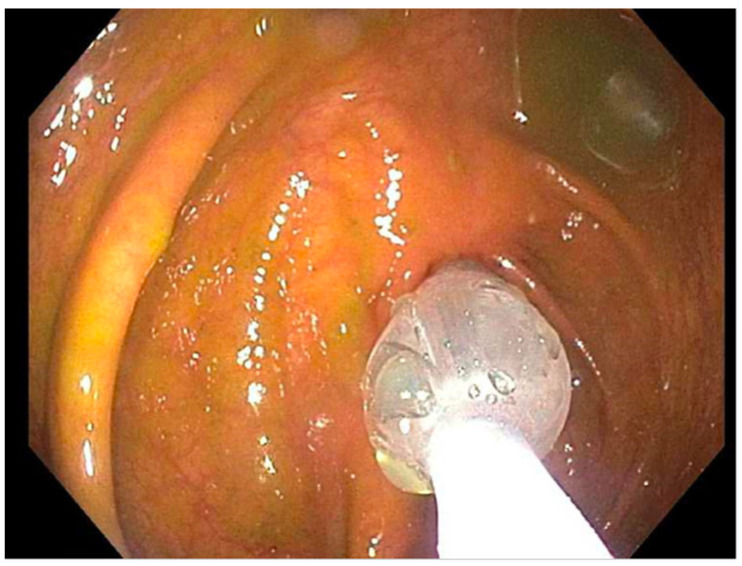
Patient 3 February 2021; dilation 7–15 mm.

**Figure 4 reports-07-00081-f004:**
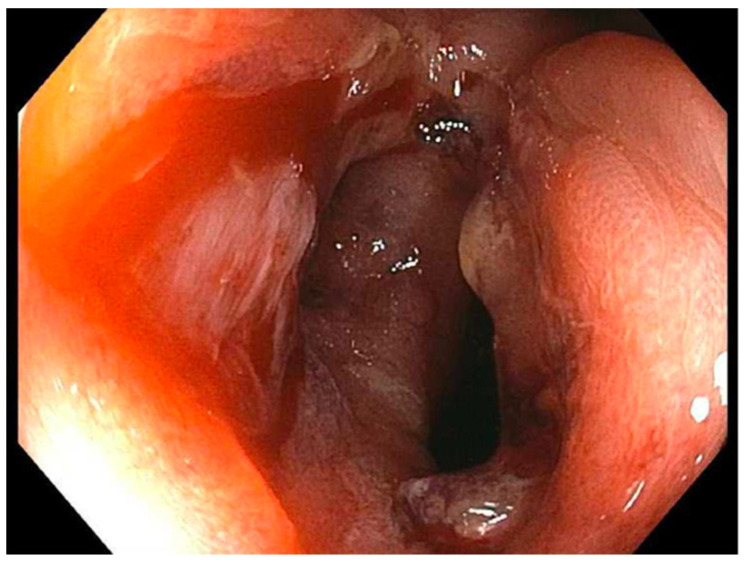
Patient 3 February 2021; 15 mm stricture post-dilation.

**Figure 5 reports-07-00081-f005:**
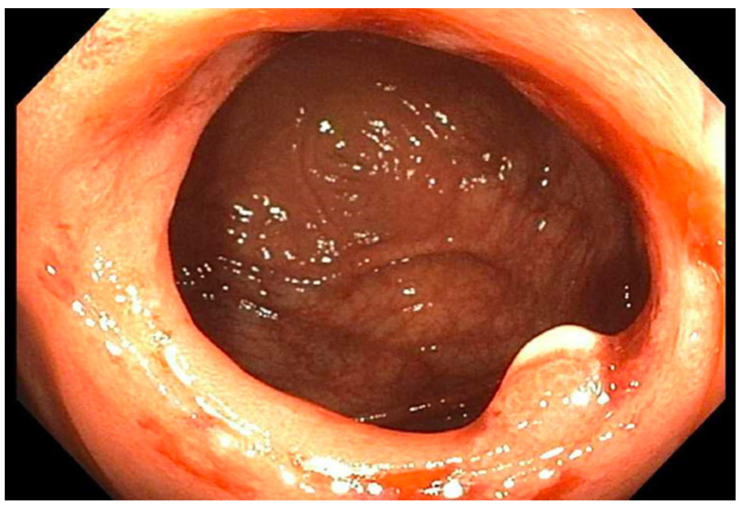
Patient 3 August 2021; dilation to 20 mm.

**Table 1 reports-07-00081-t001:** Summary of EBD procedure in CD cases.

Case	Treatment	Dilation 1	Dilation 2	Dilation 3
1	(2019: Remicade 10 mg) (2019: Humira 40 mg) (2020: Humira 80 mg subcutaneous)	November 2019, dilated (b) TI from (12 to 18 mm) (a) TI stricture dilated to (16.5 mm)	November 2020, dilated TI to (20 mm)	March 2023, laparoscopic assisted ileocecectomy
2	(2019: prednisone 40 mg) (2019: Remicade 5 mg) (Remicade 10 mg)	September 2019, ICV dilated from (8 to 11 mm)	December 2020 and August 2021 ICV dilated from (9 to 18 mm) then (12 to 18 mm)	June 2022, ICV dilated from (13.5 to 20 mm)
3	(2019: prednisone 20 mg BID w/mesalamine 500 mg TID) (2020: 6-MP 50 mg) (2021: Humira 40 mg/MTX 25 mg)	February 2021, PAC dilated from (7 to 15 mm)	August 2021, PAC dilated from (15 to 20 mm)	August 2022, mucosal narrowing stretched to 20 mm without noted renting
4	(2021: Humira 40 mg every other week/steroid 40 mg daily, escalated to Humira 40 mg weekly) (2022: MTX 25 mg)	September 2022, dilation of the ICV from (4 to 13.5 mm) (food bezoar noted)	November 2022, ICV dilated from (8 to 15 mm)	December 2022, ICV dilated (12 to 15 mm)

ICV: Ileocecal valve; TI: Terminal Ileum; a: Proximal terminal ileum; b: Distal terminal ileum; PAC: Proximal ascending colon; 6-MP: 6-mercaptopurine; MTX: Methotrexate.

**Table 2 reports-07-00081-t002:** Patient characteristics at diagnosis.

Case	Sex/Age	BMI	IBD/Dx	Stricture Site
1	F/17	60%	4/5/18	TI (2 strictures (a)/(b))
2	M/16	20%	8/30/19	ICV
3	M/18	86%	2/4/15	PAC
4	M/15	1.5%	12/22/21	ICV

ICV: Ileocecal valve; TI: Terminal Ileum; a: Proximal terminal ileum; b: Distal terminal ileum, PAC: Proximal ascending colon.

## Data Availability

The original data presented in the study are included in the article, further inquiries can be directed to the corresponding author.

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
