# Peer review of "Endoscopic Guided Dilations without Intralesional Corticosteroid Injections: Pediatric Crohn’s Patients Case Series"

_reports, 2024, doi:10.3390/reports7040081_

Round 1
Reviewer 1 Report
Comments and Suggestions for Authors
This is an interesting case series, authors demonstrated successufully EBD with favorable outcomes without the İLC injection. Article may considering for publication, ofcourse also other reviewers and editor final decisions. Minor comment: on abstract there is typo “structuring” should be “stricturing”. Reference numbers 7 and 8 seems skipped in discussion.
Good lucks!
Author Response
Thank you for the comments!
We have fixed the grammatical error in the abstract. We have also included the citations in the discussion section.
Reviewer 2 Report
Comments and Suggestions for Authors
This manuscript is a case report of 4 pediatrics patients with Crohn’s disease suffering from intestinal strictures which were successfully treated using endoscopic balloon dilation without intralesional corticosteroid injections.
This topic is interesting; however, I have serious concern in this manuscript.
Major
1. As the authors introduced, many related articles with higher evidence level have been already published although those results are conflicting. In addition, the great majority of objectives in those studies were young patients, including pediatric. Therefore, unfortunately, I found little novelty in this manuscript.
Minor
1. (P2L57-59) Description of the definition of stricture length is overlapped and mismatched.
2. I recommend that the authors provide a table which summarize patients’ characteristics, including age, gender, disease duration, stricture site, length of stricture, and so on.
3. “Crohn’s disease” should be included in the Title.
4. (P2L48-50) Please describe the results and conclusion of the study by Di Nardo et al. Furthermore, the authors should correct this reference (6).
Author Response
Thank you for your review and comments.
- Description of stricture length was fixed.
- The results and conclusion of the Di Nardo et al study were included in more detail. In addition, the reference was corrected.
- We appreciate the suggestion to include a table with patient characteristics.
- Majority of the literature published only utilized EBD with the use of intralesional corticosteroids, which is why we feel this manuscript may be of benefit to the public as we did not use ILC and they have side effects associated with them.
Reviewer 3 Report
Comments and Suggestions for Authors
Interesting work on endoscopic dilatation without corticosteroid in intestinal strictures in adolescents with Crohn’s disease.
Line 28: “ileocecal region” for ileocecal regions
Line 72: explain: RUQ/RLQ
Line 75: with Infliximab 5mg/kg. Hou many times? Once a day?
Insert a mark for every patient
Line 84: explain RLQ
Why BMI only for patient 4?
Figures cited on the text with number
Author Response
Thank you for the comments.
Abbreviations were written out. Clarification was made to dosage of Infliximab in line 75. BMI was included only for patient 4, because it was part of the presentation for the patient. However, it was removed to keep the summaries more similar.
I appreciate the recommendation to include a mark for every patient, but am unclear what is meant by that. Could you please elaborate?
Thank you!
Round 2
Reviewer 2 Report
Comments and Suggestions for Authors
The revised manuscript is improved. However, the following minor issues should be addressed.
1. The authors replied my comment as below:
Majority of the literature published only utilized EBD with the use of intralesional corticosteroids, which is why we feel this manuscript may be of benefit to the public as we did not use ILC and they have side effects associated with them.
I recommend that the authors insert this sentence in the Introduction. In addition, side effects of ILC should be concretely introduced.
2. I can’t find a table with patient characteristics.
Author Response
Comment 1: Thank you for the review. The introduction now includes side effects of ILC to help justify our reasoning behind performing dilations without them. It also includes the statement about previous literature focusing on dilations with ILC with little evidence without.
Comment 2: Two tables with patient characteristics have now been added right before the discussion section.